# Classification of Complex Fuzzy Numbers and Fuzzy Inner Products

**Jin Hee Yoon †, Taechang Byun \*,†, Ji Eun Lee †**  **and Keun Young Lee †**

Department of Mathematics and Statistics, Sejong University, Seoul 147-747, Korea; jin9135@sejong.ac.kr (J.H.Y.); jieunlee7@sejong.ac.kr (J.E.L.); bst21@sejong.ac.kr (K.Y.L.)
\* Correspondence: tcbyun@sejong.ac.kr
† These authors contributed equally to this work.

**Abstract:** The paper is concerned with complex fuzzy numbers and complex fuzzy inner product spaces. In the classical complex number set, a complex number can be expressed using the Cartesian form or polar form. Both expressions are needed because one expression is better than the other depending on the situation. Likewise, the Cartesian form and the polar form can be defined in a complex fuzzy number set. First, the complex fuzzy numbers (CFNs) are categorized into two types, the polar form and the Cartesian form, as type I and type II. The properties of the complex fuzzy number set of those two expressions are discussed, and how the expressions can be used practically is shown through an example. Second, we study the complex fuzzy inner product structure in each category and find the non-existence of an inner product on CFNs of type I. Several properties of the fuzzy inner product space for type II are proposed from the modulus that is newly defined. Specfically, the Cauchy-Schwartz inequality for type II is proven in a compact way, not only the one for fuzzy real numbers. In fact, it was already discussed by Hasanhani et al; however, they proved every case in a very complicated way. In this paper, we prove the Cauchy-Schwartz inequality in a much simpler way from a general point of view. Finally, we introduce a complex fuzzy scalar product for the generalization of a complex fuzzy inner product and propose to study the condition for its existence on CFNs of type I.

**Keywords:** complex fuzzy numbers; modulus; inner product; scalar product; complex fuzzy inner product space

**MSC:** Primary 54A40; 03E72; Secondary 46Cxx

## 1. Introduction

In classical complex analysis, we have two different types of notations in the complex number set $\mathbb{C}$. One is the Cartesian form ($z = a + bi$), and the other one is the polar form ($z = re^i\theta$). Even though we have the Cartesian form, the polar form is needed depending on the situation. The Cartesian form is generally convenient, but if we have to use the angle such as in triangular functions, the polar form is better. Likewise, in fuzzy complex analysis ($CF(\mathbb{C})$ or $CF(\mathbb{R})$; see Section 3), we can first consider the Cartesian complex fuzzy number form, but depending on the situation, it is necessary to define the polar form to express some special cases (see Example 2). Sometimes, the polar form in $CF(\mathbb{C})$ or $CF(\mathbb{R})$ is better than the Cartesian form. For example, if we want to express some situation with a "periodic" time series that has a fuzzy amplitude, the polar form is better than the Cartesian form. As we see in Example 2, we can express the periodicity easily if the data include the monthly property or the seasonality. However, due to the complexity of $CF(\mathbb{C})$ or $CF(\mathbb{R})$, not all the properties in $\mathbb{C}$ hold in $CF(\mathbb{C})$ or $CF(\mathbb{R})$. Therefore, in this paper, we discuss similar properties in $\mathbb{C}$ that we can even

have in $CF(\mathbb{C})$ or $CF(\mathbb{R})$, and we also discuss some properties that we cannot have in $CF(\mathbb{C})$ or $CF(\mathbb{R})$. For example, as we discuss in the paper, the complex fuzzy inner product does not exist based on the polar form. The Cauchy-Schwartz inequality holds using the Cartesian form in the complex fuzzy number set.

The definitions of complex fuzzy numbers have been introduced in several studies [1–3]. The complex fuzzy numbers can be applied to many real applications [3]. Buckley [1] first introduced a complex number approach to fuzzy numbers, which was named "fuzzy complex numbers". He defined the Cartesian form and the polar form of fuzzy complex numbers $z$ based on their membership function, which is a mapping from the complex numbers into $[0,1]$. On the other hand, Ramot et al. [3] introduced a complex-valued grade of fuzzy membership functions to the magnitude (modulus) and argument of the polar form to define complex fuzzy numbers. Fu and Shen [2] introduced a complex-valued grade of fuzzy membership functions to the real part and the imaginary part to define fuzzy complex numbers. There were some studies that dealt with the fuzzy inner product [4–6] and the fuzzy Hilbert space [7,8]. Recently, we [9] discussed the absence of non-trivial fuzzy inner product spaces and the Cauchy-Schwartz inequality in the fuzzy real number system.

In this paper, we discuss more properties of the complex fuzzy numbers as an extension of the previous study [1]. This approach is different from Buckley's [1]. We deal with complex fuzzy numbers, not "fuzzy complex numbers".

In Sections 3 and 4, the complex fuzzy numbers (CFNs) are categorized into two types, as CFNs of type I for the polar form and CFNs of type II for the Cartesian form. The basic properties of the operations in each category are checked.

A fuzzy Hilbert space has been introduced in [7,8]. This paper suggests a complex fuzzy inner product space based on our new approach. In Section 5, we discuss the complex fuzzy inner product on each category of CFN, which shows that, unlike the complex number, the complex fuzzy number of the polar form (type I) has no relation to that of the Cartesian form (type II). In fact, the non-existence of inner products on CFN of type I is proved. We introduce the definition of a complex fuzzy inner product space for type II. Also, several properties of the fuzzy inner product space for type II have been proposed from the modulus defined in Section 5. Especially, the Cauchy-Schwartz inequality for type II is proved in in a compact way, not only the one for fuzzy real numbers. In fact, it was already dealt with in [7], however they proved every case in a very complicated way. In this paper, we prove the Cauchy-Schwartz inequality in a much simpler way in a general point of view.

In Section 6, we introduce a complex fuzzy scalar product and investigate its properties.

## 2. Preliminaries

In this section, we provide basic definitions and notations for this study.

**Definition 1** ([10] (p. 390)). *A mapping $\eta : \mathbb{R} \to [0,1]$ is called a fuzzy real number with $\alpha$-level set $[\eta]_\alpha = \{t : \eta(t) \geqslant \alpha\}$, if it satisfies the following conditions:*
*(i) there exists $t_0 \in \mathbb{R}$ such that $\eta(t_0) = 1$.*
*(ii) for each $\alpha \in (0,1]$, there exist real numbers $\eta_\alpha^- \leqslant \eta_\alpha^+$ such that the $\alpha$-level set $[\eta]_\alpha$ is equal to the closed interval $[\eta_\alpha^-, \eta_\alpha^+]$.*

**Remark 1.** *The condition (ii) of Definition 1 is equivalent to convex and upper semi continuous:*
*(1) a fuzzy real number $\eta$ is convex if $\eta(t) \geq \min\{\eta(s), \eta(r)\}$ where $s \leq t \leq r$.*
*(2) a fuzzy real number $\eta$ is called upper semi-continuous if for all $t \in \mathbb{R}$ and $\epsilon > 0$ with $\eta(t) = \alpha$, there is $c > 0$ such that $|s - t| < c = c(t) \Rightarrow \eta(t) < a + \epsilon$, i.e., $\eta^{-1}([0, a + \epsilon))$ for all $a \in [0,1]$ and $\epsilon > 0$ is open in the usual topology of $\mathbb{R}$.*

The set of all fuzzy real numbers is denoted by $F(\mathbb{R})$. If $\eta \in F(\mathbb{R})$ and $\eta(t) = 0$ whenever $t < 0$, then $\eta$ is called a non-negative fuzzy real number and $F^*(\mathbb{R})$ denotes the set of all non-negative fuzzy

real numbers. We note that real number $\eta_\alpha^- \geqslant 0$ for all $\eta \in F^*(\mathbb{R})$ and all $\alpha \in (0,1]$. Each $r \in \mathbb{R}$ can be considered as the fuzzy real number $\tilde{r} \in F(\mathbb{R})$ denoted by

$$\tilde{r}(t) = \begin{cases} 1, & t = r, \\ 0, & t \neq r, \end{cases}$$

hence it follows that $\mathbb{R}$ can be embedded in $F(\mathbb{R})$.

**Definition 2** ([11] (p. 216))**.** *The arithmetic operations* $\oplus, \ominus, \otimes$ *and* $\oslash$ *on* $F(\mathbb{R}) \times F(\mathbb{R})$ *are defined by*

$$(\eta \oplus \gamma)(t) = \sup_{t=x+y} (\min(\eta(x), \gamma(y))),$$

$$(\eta \ominus \gamma)(t) = \sup_{t=x-y} (\min(\eta(x), \gamma(y))),$$

$$(\eta \otimes \gamma)(t) = \sup_{t=xy} (\min(\eta(x), \gamma(y))),$$

$$(\eta \oslash \gamma)(t) = \sup_{t=x/y} (\min(\eta(x), \gamma(y))),$$

*which are special cases of Zadeh's extension principles.*

**Definition 3** ([11] (p. 216, Equation (2.7)))**.** *The absolute value* $|\eta|$ *of* $\eta \in F(\mathbb{R})$ *is defined by*

$$|\eta|(t) = \begin{cases} \max(\eta(t), \eta(-t)), & t \geqslant 0, \\ 0, & t < 0. \end{cases}$$

**Lemma 1** ([11] (p. 217))**.** *Let* $\eta, \gamma \in F(\mathbb{R})$ *and* $[\eta]_\alpha = [\eta_\alpha^-, \eta_\alpha^+], [\gamma_\alpha] = [\gamma_\alpha^-, \gamma_\alpha^+]$. *Then for all* $\alpha \in (0,1]$,

$$\begin{aligned}
[\eta \oplus \gamma]_\alpha &= [\eta_\alpha^- + \gamma_\alpha^-, \eta_\alpha^+ + \gamma_\alpha^+], \\
[\eta \ominus \gamma]_\alpha &= [\eta_\alpha^- - \gamma_\alpha^+, \eta_\alpha^+ - \gamma_\alpha^-], \\
[\eta \otimes \gamma]_\alpha &= [\eta_\alpha^- \gamma_\alpha^-, \eta_\alpha^+ \gamma_\alpha^+], \forall \eta, \gamma \in F^*(\mathbb{R}), \\
[\tilde{1} \oslash \eta]_\alpha &= [\frac{1}{\eta_\alpha^+}, \frac{1}{\eta_\alpha^-}], \forall \eta_\alpha^- > 0, \\
[|\eta|]_\alpha &= [\max(0, \eta_\alpha^-, -\eta_\alpha^+), \max(|\eta_\alpha^-|, |\eta_\alpha^+|)].
\end{aligned}$$

**Definition 4** ([11])**.** *Let* $\eta, \gamma \in F(\mathbb{R})$ *and* $[\eta]_\alpha = [\eta_\alpha^-, \eta_\alpha^+], [\gamma_\alpha]_\alpha = [\gamma_\alpha^-, \gamma_\alpha^+]$, *for all* $\alpha \in (0,1]$. *Define a partial ordering by* $\eta \leq \gamma$ *in* $F(\mathbb{R})$ *if and only if* $\eta_\alpha^- \leqslant \gamma_\alpha^-, \eta_\alpha^+ \leqslant \gamma_\alpha^+$, *for all* $\alpha \in (0,1]$. *The strict inequality in* $F(\mathbb{R})$ *is defined by* $\eta < \gamma$ *if and only if* $\eta_\alpha^- < \gamma_\alpha^-, \eta_\alpha^+ < \gamma_\alpha^+$, *for all* $\alpha \in (0,1]$.

## 3. Complex Fuzzy Numbers of Type I

Recall that the polar representation $z = re^{i\theta}$ of a complex number, where $r \geq 0$ and $\theta \in \mathbb{R}$ [3]. In this section, we consider its extension in fuzzy category, called complex fuzzy numbers of type I, and investigate their basic properties under some structure.

**Definition 5** ([3] (p. 171))**.** *A complex fuzzy set S of type I defined on a universe of discourse U, is characterized by a membership function*

$$\mu_S(x) := r_S(x) \cdot e^{i\theta_S(x)} (\text{ for any } x \in U)$$

*that assigns a complex-valued grade of membership in S where* $r_S(x)$ *and* $\theta_S(x)$ *are both real-valued and* $r_S(x)$ *is a fuzzy real number on U. Here,* $r_S(x) = |\mu_S(x)|$ *is called the amplitude of* $\mu_S(x)$.

The complex fuzzy set $S$ may be represented as the set of *ordered pairs*

$$S = \{(x, \mu_S(x)) : x \in U\}. \tag{1}$$

Set $D^2 := \{re^{i\theta} \mid 0 \leq r \leq 1 \text{ and } 0 \leq \theta < 2\pi\}$. We now give examples of a complex fuzzy set $S$ of type I.

**Example 1.** *Let us consider a complex fuzzy set*

$$S = \{(x, \mu_S(x)) : x \in \mathbb{R}\} \subset \mathbb{R} \times D^2, \tag{2}$$

*defined by*

$$\mu_S(x) = r_S(x) \cdot e^{i\theta_S(x)} = \begin{cases} (1 - \frac{x^2}{100})e^{2\pi i x}, & |x| < 10, \\ 0, & |x| \geq 10. \end{cases}$$

*Then $\mu_S(x)$ can be expressed in Figure 1.*

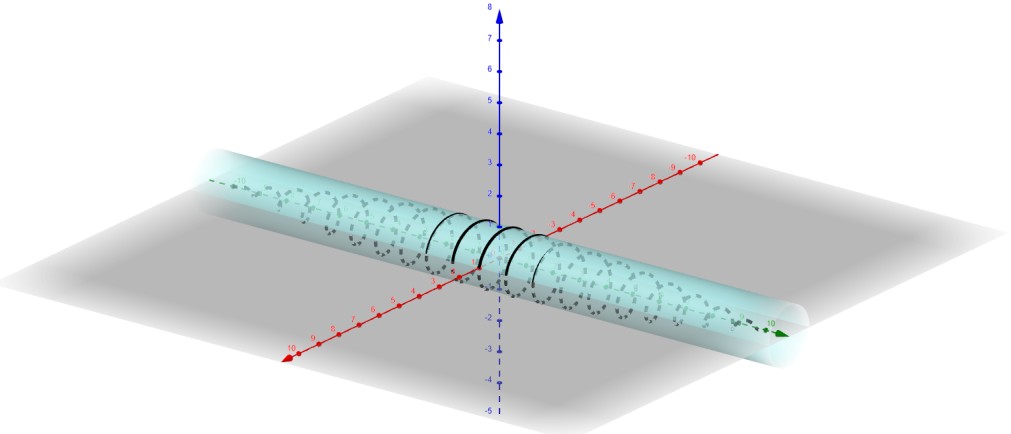

**Figure 1.** The complex fuzzy number of type I in Example 1 .

**Example 2** ([3] (p. 184, Equation (49))). *Let $U$ be the set of financial indicators or indexes of the American economy. Possible elements of this set are unemployment rate, inflation, interest rates, growth rate, GDP, Dow-Jones industrial average, etc. Let $V$ be the set of financial indicators of the Japanese Economy. Let the complex fuzzy relation $R(U, V)$ represent the relation of influence of American financial indexes on Japanese financial indexes: y is influenced by x", where $x \in U$ and $y \in V$. The membership function for the relation $R(U, V)$ and $\mu_R(x, y)$, can be presented by complex valued, with an amplitude term and a phase term. The amplitude term indicates the degree of influence of an American financial index on a Japanese financial index. Consider, for example, $\mu_R(Growth\ rate, export)$, i.e., the grade of membership associated with the statement: "American growth rate influences Japanese Export". Assume the interactions between American and Japanese financial indicators are measured in the limited time frame of 12 months, then it can be represented by*

$$\mu_R(Growth\ rate, export) = 0.8 \cdot e^{\frac{4}{12} 2\pi \cdot i}.$$

*Note that the amplitude term was selected to be 0.8, similar to the grade of membership of a traditional fuzzy set. The phase term was chosen to be $\frac{4}{12} \cdot 2\pi$ as an average of three-five months, normalized by 12 months.*

Consider the case both $S = \mathbb{C}$ and $\mathbb{R}$ respectively, which given $CF_I(\mathbb{C})$ and $CF_I(\mathbb{R})$.

**Definition 6.** *A mapping* $\eta : \mathbb{C} \to D^2$ *(or* $\eta : \mathbb{R} \to D^2$, *respectively) is called a complex fuzzy number (CFN) on* $\mathbb{C}$ *( or* $\mathbb{R}$*), respectively, whose* $\alpha$*-level set is denoted by*

$$[\eta]_\alpha = \{z \in \mathbb{C} : |\eta(z)| \geq \alpha\},$$

*if it satisfies two axioms;*
*(i) There exists* $z_0 \in \mathbb{C}$ *( or* $\mathbb{R}$*), respectively, such that* $|\eta(z_0)| = 1$.
*(ii) For each* $\alpha \in (0,1]$, $[\eta]_\alpha$ *is a compact connected convex set in* $\mathbb{C}$.

Note that if $\eta \in CF_I(\mathbb{R})$, then there exists real numbers $-\infty < \eta_\alpha^- \leq \eta_\alpha^+ < +\infty$ such that $[\eta]_\alpha = [\eta_\alpha^-, \eta_\alpha^+]$.

It is obvious that both $\mathbb{R}$ and $F(\mathbb{R})$ can be embedded in $CF_I(\mathbb{R})$. Furthermore, both $\mathbb{C}$ and $CF_I(\mathbb{R})$ can be embedded in $CF_I(\mathbb{C})$. In this section, $CF_I(\mathbb{R})$ and $CF_I(\mathbb{C})$ will be written by $CF(\mathbb{R})$ and $CF(\mathbb{C})$ for the simplicity, respectively.

**Example 3.** *Let*

$$\eta(t) = \begin{cases} \frac{1}{|t|} e^{2\pi t i}, & |t| > 1, \\ 1 \cdot e^{2\pi t i}, & |t| \leq 1, \end{cases}$$

*where* $\eta \in CF(\mathbb{R})$. *Then* $[\eta(t)]_{\frac{1}{2}} = [-2, 2]$ *for all* $t \in \mathbb{R}$.

**Example 4.** *Let* $z = re^{i\theta}$ *and* $\alpha \in (0,1]$. *Define*

$$\eta(re^{i\theta}) = \begin{cases} \frac{1}{r} e^{2\pi \theta i}, & r \geq 1, \\ 1 \cdot e^{2\pi \theta i}, & 0 \leq r < 1, \end{cases}$$

*where* $\eta \in CF(\mathbb{C})$. *Then* $[\eta]_\alpha = \{z \in \mathbb{C} : |z| \leq \frac{1}{\alpha}\}$.

Given $r \in \mathbb{R}$, recall that the definition of $\widetilde{r} \in F(\mathbb{R})$ given by

$$\widetilde{r}(t) = \begin{cases} 1, & t = r, \\ 0, & t \neq r. \end{cases}$$

Let $CF(U)$ be the set of all *CFN*s on $U = \mathbb{R}$ or $\mathbb{C}$. Note that each $z = re^{i\theta} \in \mathbb{C}$ can be considered as the *CFN* $\widetilde{z} \in CF(\mathbb{C})$ defined by

$$\widetilde{z}(t) = \begin{cases} e^{i\theta}, & t = r, \\ 0, & t \neq r. \end{cases} \tag{3}$$

Then, $\widetilde{z} \in CF(U)$ is the extension of $\widetilde{r} \in F(\mathbb{R})$. Moreover, $\mathbb{C}$ can be embedded in $CF(\mathbb{C})$.
Since each $z \in \mathbb{R}$ can be considered as the complex fuzzy number $\widehat{z} \in CF(\mathbb{C})$ defined by

$$\widehat{z}(t) = |\widetilde{z}(t)| = \begin{cases} 1, & t = r, \\ 0, & t \neq r. \end{cases} \tag{4}$$

it follows that $\mathbb{R}$ can be embedded in $CF(\mathbb{C})$.

In this paper, we focus on *CFN* based on $\mathbb{R}$.

**Definition 7.** *Let* $\eta(t) = r_\eta(t) e^{i\theta_\eta(t)}$ *and* $\delta(t) = r_\delta(t) e^{i\theta_\delta(t)}$ *be complex fuzzy numbers on* $\mathbb{R}$. *Then*

$$\eta(t) = \delta(t) \text{ for all } t \in \mathbb{R} \text{ if } r_\eta(t) \equiv r_\delta(t) \text{ and } \theta_\eta(t) \equiv \theta_\delta(t) \ (mod \ 2\pi)$$

*for all $\alpha \in (0,1]$.*

Note that $r_\eta(t) \equiv r_\delta(t)$ means $[r_\eta]_\alpha \equiv [r_\delta]_\alpha$ for all $\alpha \in [0,1]$.

Note that $\eta(t) \in \mathbb{C}$ for each $t \in \mathbb{R}$ so that $|\eta(t)| \in \mathbb{R}$. As the square root of a polar form of a complex number, that of a complex fuzzy number is done as follows:

**Definition 8.** *For a CFN $\eta(t) = r_\eta(t)e^{i\theta(t)}$ on $\mathbb{R}$, we define the square root of $\eta(t)$ by*

$$\sqrt{\eta(t)} = \sqrt{r_\eta(t)}e^{i\frac{\theta(t)}{2}} \in CF(\mathbb{R})$$

*where $r(t)$ is the amplitude of $\eta(t)$ and defined by $r_\eta(t) = |\eta(t)|$.*

**Definition 9.** *The absolute value $|\eta|$ of $\eta \in CF(\mathbb{R})$ of type I is defined by*

$$|\eta|(t) = \begin{cases} max(|\eta(t)|, |\eta(-t)|), & t \geq 0, \\ 0, & t < 0. \end{cases}$$

The Definition 9 implies $|\eta| \in F(\mathbb{R})$.

Based on [3] (p. 181), Definitions 10 and 11 are suggested.

**Definition 10.** *The arithmetic operation $\oplus$ is given by*

$$(\eta \oplus \delta)(t) := \sup_{t=x+y} \{min(|\eta(x)|, |\delta(y)|)\} \cdot e^{i\theta_{\eta \oplus \delta}},$$

*where $\theta_{\eta \oplus \delta}$ can be defined by $\theta_{\eta \oplus \delta}(t) = \theta_\eta(t) + \theta_\delta(t)$.*

Remark that, in Definition 10, $\theta_{\eta \oplus \delta}$ can be defined in several ways as follows;

(i) $\theta_{\eta \oplus \delta}(t) = max(\theta_\eta(t), \theta_\delta(t))$,

(ii) $\theta_{\eta \oplus \delta}(t) = min(\theta_\eta(t), \theta_\delta(t))$,

(iii) $\theta_{\eta \oplus \delta}(t) = \begin{cases} \theta_\eta(t), & \text{if } |\eta(t)| \geq |\delta(t)|, \\ \theta_\delta(t), & \text{if } |\eta(t)| < |\delta(t)|. \end{cases}$

Similarly, we can think the following definitions in may cases related to angle terms.

**Definition 11.** *(i) The arithmetic operation $\ominus$ is given by*

$$(\eta \ominus \delta)(t) := \sup_{t=x-y} \{min(|\eta(x)|, |\delta(y)|)\} \cdot e^{i\theta_{\eta \ominus \delta}},$$

*where $\theta_{\eta \ominus \delta}$ can be defined by $\theta_{\eta \ominus \delta}(t) = \theta_\eta(t) - \theta_\delta(t)$.*

*(ii) The arithmetic operation $\otimes$ is given by*

$$(\eta \otimes \delta)(t) := \sup_{t=xy} \{min(|\eta(x)|, |\delta(y)|)\} \cdot e^{i\theta_{\eta \otimes \delta}},$$

*where $\theta_{\eta \otimes \delta}$ can be defined by $\theta_{\eta \otimes \delta}(t) = \theta_\eta(t) + \theta_\delta(t)$.*

*(iii) The arithmetic operation $\oslash$ is given by*

$$(\eta \oslash \delta)(t) := \sup_{t=x/y} \{min(|\eta(x)|, |\delta(y)|)\} \cdot e^{i\theta_{\eta \oslash \delta}},$$

*where $\theta_{\eta \oslash \delta}$ can be defined by $\theta_{\eta \oslash \delta}(t) = \theta_\eta(t) - \theta_\delta(t)$.*

**Lemma 2.** *Let $\eta, \delta \in CF(\mathbb{R})$, $[\eta]_\alpha = [\eta_\alpha^-, \eta_\alpha^+]$, and $[\delta]_\alpha = [\delta_\alpha^-, \delta_\alpha^+]$. Then for all $\alpha \in (0, 1]$,*
*(i) $[\eta \oplus \delta]_\alpha = [\eta_\alpha^- + \delta_\alpha^-, \eta_\alpha^+ + \delta_\alpha^+]$,*
*(ii) $[\eta \ominus \delta]_\alpha = [\eta_\alpha^- - \delta_\alpha^+, \eta_\alpha^+ - \delta_\alpha^-]$,*
*(iii) $[\eta \otimes \delta]_\alpha = [\eta_\alpha^- \delta_\alpha^-, \eta_\alpha^+ \delta_\alpha^+]$,*
*(iv) $[\eta \oslash \delta]_\alpha = [\frac{\eta_\alpha^-}{\delta_\alpha^+}, \frac{\eta_\alpha^+}{\delta_\alpha^-}]$ if $\eta_\alpha^- > 0, \delta_\alpha^- > 0$,*
*(v) $[|\eta|]_\alpha = [\max(0, \eta_\alpha^-, -\eta_\alpha^+), \max(|\eta_\alpha^-|, |\eta_\alpha^+|)]$.*

## 4. Complex Fuzzy Numbers of Type II

Note that each $z = a + bi \in \mathbb{C}$ can be considered as the *CFN* of type II, $\tilde{z} \in CF_{II}(\mathbb{C})$, defined by

$$\tilde{z} = \tilde{a} \oplus i\,\tilde{b}, \tag{5}$$

where " i " is the *imaginary unit* of a complex fuzzy number of type II, ( not of a complex number). Then, $\tilde{z} \in CF_{II}(\mathbb{C})$ is the extension of $\tilde{r} \in F(\mathbb{R})$. Moreover, $\mathbb{C}$ can be embedded in $CF(\mathbb{C})$. In this section, $CF_{II}(\mathbb{C})$ will be written by $CF(\mathbb{C})$ for the simplicity.

**Definition 12.** *A complex fuzzy set S of type II defined on a universe of discourse U, is characterized by a membership function*
$$\mu_S(t) = (x_S + i\,y_S)(t) := x_S(t) + i\,y_S(t) \text{ for any } t \in U$$
*that assigns a complex-valued grade of fuzzy membership in S where both $x_S$ and $y_S$ are fuzzy real numbers on S.*

**Definition 13** ([2] (p. 1405))**.** *Let $z_1 = a \oplus i\,b$ and $z_2 = c \oplus i\,d$ be complex fuzzy numbers where $a, b, c$, and $d$ are fuzzy real numbers. The basic arithmetic operations on $z_1$ and $z_2$ are defined as follows;*
*(i) Addition: $z_1 \oplus z_2 = (a \oplus c) \oplus i\,(b \oplus d)$,*
*(ii) Subtraction: $z_1 \ominus z_2 = (a \ominus c) \oplus i\,(b \ominus d)$,*
*(iii) Multiplication: $z_1 \otimes z_2 = (ac \ominus bd) \oplus i\,(bc \oplus ad)$,*
*(iv) Division: $z_1 \oslash z_2 = (\frac{ac \oplus bd}{c^2 \oplus d^2}) \oplus i\,(\frac{bc \ominus ad}{c^2 \oplus d^2})$,*
*(v) Conjugate: $\overline{z_1} = a \oplus i\,(\ominus b)$.*

**Remark 2.** *In [2] (p. 1405), for $z = a + ib$, the definition on the modulus of complex fuzzy number $z \in CF(\mathbb{R})$ of type II is given by*
$$|z| = \sqrt{|a|^2 \oplus |b|^2}$$
*which satisfies $|z|(t) = \sup_{t = \sqrt{m^2 + n^2}}(\min\{a(m), b(n)\})$. However, given a fuzzy real number $\xi \in F(\mathbb{R})$, $\xi \otimes \xi$ need not be greater than or equal to $\tilde{0}$. For example, let $\xi \in F(\mathbb{R})$ be defined by*

$$\xi(t) = \begin{cases} 1 + t, & -1 \leq t \leq 0 \\ 1 - t, & 0 \leq t \leq 1 \\ 0, & \text{otherwise.} \end{cases}$$

*Then*

$$\begin{aligned} (\xi \otimes \xi)(-\frac{1}{2}) &= \sup_{st = -\frac{1}{2}} \min\{\xi(t), \xi(s)\} \\ &\geq \min\{\xi(\frac{1}{\sqrt{2}}), \xi(-\frac{1}{\sqrt{2}})\} = 1 - \frac{1}{\sqrt{2}}. \end{aligned}$$

*Hence $(\xi \otimes \xi)_{1 - \frac{1}{\sqrt{2}}}^- < -\frac{1}{2} < 0$.*

*Thus the definition on the modulus might need to be changed as follows:*

**Definition 14.** *The modulus of a complex fuzzy number $\xi \in CF(\mathbb{R})$ of type II is*

$$|\xi| = \sqrt{\left(|Re\xi| \otimes |Re\xi|\right) \oplus \left(|Im\xi| \otimes |Im\xi|\right)}.$$

Note that the modulus of the Definition 14 is different from the absolute value of the Definition 9 even though they are equivalent in $\mathbb{C}$. The triangle inequality on modulus may be shown in Theorem 1 with respect to type II.

**Theorem 1.** *Given two fuzzy real numbers $\xi, \eta \in CF(\mathbb{R})$, it holds that*

$$|\xi \oplus \eta| \leq |\xi| \oplus |\eta|.$$

**Proof.** Let $\alpha \in (0,1]$ be given. Recall that for any fuzzy real number $\zeta \in F(\mathbb{R})$ with $\zeta \geq \tilde{0}$, it holds that $(\sqrt{\zeta})_\alpha^- = \sqrt{\zeta_\alpha^-}$. Then

$$
\begin{aligned}
(|\xi| \oplus |\eta|)_\alpha^- &= \left(\sqrt{|Re\xi|^2 \oplus |Im\xi|^2} \oplus \sqrt{|Re\eta|^2 \oplus |Im\eta|^2}\right)_\alpha^- \\
&= \sqrt{(|Re\xi|_\alpha^-)^2 + (|Im\xi|_\alpha^-)^2} + \sqrt{(|Re\eta|_\alpha^-)^2 + (|Im\eta|_\alpha^-)^2} \\
&\geq \sqrt{(|Re\xi|_\alpha^- + |Re\eta|_\alpha^-)^2 + (|Im\xi|_\alpha^- + |Im\eta|_\alpha^-)^2} \\
&\geq \sqrt{(|Re\xi \oplus Re\eta|_\alpha^-)^2 + (|Im\xi \oplus Im\eta|_\alpha^-)^2} \\
&= \sqrt{(|Re(\xi \oplus \eta)|_\alpha^-)^2 + (|Im(\xi \oplus \eta)|_\alpha^-)^2} \\
&= \left(\sqrt{|Re(\xi \oplus \eta)|^2 \oplus |Im(\xi \oplus \eta)|^2}\right)_\alpha^- \\
&= |\xi \oplus \eta|_\alpha^-.
\end{aligned}
$$

Similarly, we get $(|\xi| \oplus |\eta|)_\alpha^+ \geq |\xi \oplus \eta|_\alpha^+$.  $\square$

## 5. Complex Fuzzy Inner Product Space

In this section, we introduce the definition and investigate some properties of a complex fuzzy inner product space for type II. We will use operations in [2] (p. 1405).

**Definition 15.** *Let $X$ be a vector space over $\mathbb{F}(\mathbb{R}$ or $\mathbb{C})$. Assume the mappings $L, R : [0,1] \times [0,1] \to [0,1]$ are symmetric and non-decreasing in both arguments, and that $L(0,0) = 0$ and $R(1,1) = 1$. Let $\|\cdot\| : X \to F^*(\mathbb{R})$. The quadruple $(X, \|\cdot\|, L, R)$ is called a fuzzy normed space [12] with the fuzzy norm $\|\cdot\|$, if the following conditions are satisfied:*
*(F1) if $x \neq 0$, then $\inf_{0 < \alpha \leq 1} \|x\|_\alpha^- > 0$,*
*(F2) $\|x\| = \tilde{0}$ if and only if $x = 0$,*
*(F3) $\|rx\| = |\tilde{r}|\|x\|$ for $x \in X$ and $r \in \mathbb{R}$,*
*(F4) for all $x, y \in X$,*
  *(F4L) $\|x + y\|(s + t) \geq L(\|x\|(s), \|y\|(t))$ whenever $s \leq \|x\|_1^-, t \leq \|y\|_1^-$ and $s + t \leq \|x + y\|_1^-$,*
  *(F4R) $\|x + y\|(s + t) \leq R(\|x\|(s), \|y\|(t))$ whenever $s \geq \|x\|_1^-, t \geq \|y\|_1^-$ and $s + t \geq \|x + y\|_1^-$.*

Here, we fix $L(s,t) = \min(s,t)$ and $R(s,t) = \max(s,t)$ for all $s, t \in [0,1]$ and we write $(X, \|\cdot\|)$.

**Definition 16.** *Let $X$ be a vector space over $\mathbb{C}$. A complex-valued fuzzy inner product on $X$ is a mapping $\langle \, , \, \rangle : X \times X \to CF(\mathbb{R})$ such that for all vectors $x, y, z \in X$ ans $r \in \mathbb{C}$, we have*
*$(IP_1)$ $\langle x + y, z \rangle = \langle x, z \rangle \oplus \langle y, z \rangle$,*

$(IP_2)$ $\langle rx, y \rangle = \widetilde{r} \otimes \langle x, y \rangle$,

$(IP_3)$ $\langle x, y \rangle = \overline{\langle y, x \rangle}$,

$(IP_4)$ $\langle x, x \rangle \geq \widetilde{0}$,

$(IP_5)$ $\inf_{0 < \alpha \leq 1} \langle x, x \rangle_\alpha^- > 0$, if $x \neq 0$,

$(IP_6)$ $\langle x, x \rangle = \widetilde{0}$ if and only if $x = 0$.

The vector space $X$ with a complex-valued fuzzy inner product is called a complex fuzzy inner product space.

*5.1. Non-Existence of the Inner Product on CFNs of Type I*

In this subsection, a complex fuzzy inner product in view of type I can not be defined. Let $V$ be a given complex $n$-dimensional vector space. To show by a contradiction, assume that $\langle \cdot, \cdot \rangle : V \times V \to CF(\mathbb{R})$ is a complex fuzzy inner product on $V$. It is well known that there is a basis $\beta = \{v_1, \cdots, v_n, iv_1, \cdots, iv_n\}$ on a real vector space $V$. Let $x = v_1$ and $y = iv_1$. Recall that, for $z = re^{i\theta}$, the *CFN* $\widetilde{z} \in CF(\mathbb{R})$ is given by

$$\widetilde{z}(t) = \begin{cases} e^{i\theta}, & t = r, \\ 0, & t \neq r. \end{cases} \tag{6}$$

**Lemma 3.** *If $r > 0$, then $\widetilde{re^{i\theta}} = \widetilde{r}e^{i\phi}$ for some function $\phi : \mathbb{R} \to \mathbb{R}$ satisfying $e^{i\phi(r)} = e^{i\theta}$.*

**Proof.** Assume $r > 0$ and let $\widetilde{re^{i\theta}} = fe^{i\phi}$ for some fuzzy real number $f \in F(\mathbb{R})$ and for a function $\phi : \mathbb{R} \to \mathbb{R}$. Then, Equation (6) says that, for $t \neq r$,

$$f(t)e^{i\phi(t)} = \widetilde{re^{i\theta}}(t) = 0,$$

which implies that $f(t) = 0$ for $t \neq r$. For $t = r$, it also does that

$$f(r)e^{i\phi(r)} = e^{i\theta},$$

which gives both $f(r) = 1$ and $e^{i\phi(r)} = e^{i\theta}$. Thus we get $f = \widetilde{r}$. $\square$

The following example shows that the inner product of CFN of type I on a vector space cannot be defined.

**Example 5.** *Let $x = v_1$ and $y = iv_1$. From the Lemma 3, there are two functions $\phi, \psi : \mathbb{R} \to \mathbb{R}$ such that $\widetilde{\sqrt{2}e^{i\pi/4}} = \sqrt{2}e^{i\phi}$ and $\widetilde{e^{i\pi/2}} = \widetilde{1}e^{i\psi}$ satisfying $e^{i\phi(\sqrt{2})} = e^{i\pi/4}$ and $e^{i\psi(1)} = e^{i\pi/2}$. Then*

$$\begin{aligned}
\langle x + y, v_1 \rangle &= \langle \sqrt{2}e^{i\pi/4}v_1, v_1 \rangle \\
&= \widetilde{\sqrt{2}e^{i\pi/4}} \otimes \langle v_1, v_1 \rangle \\
&= \left( \widetilde{\sqrt{2}} \otimes \langle v_1, v_1 \rangle \right) e^{i\phi}
\end{aligned}$$

*and*

$$\begin{aligned}
\langle x, v_1 \rangle \oplus \langle y, v_1 \rangle &= \langle v_1, v_1 \rangle \oplus \left( \widetilde{e^{i\pi/2}} \otimes \langle v_1, v_1 \rangle \right) \\
&= \langle v_1, v_1 \rangle \oplus \left( (\widetilde{1} \otimes \langle v_1, v_1 \rangle) e^{i\psi} \right) \\
&= \langle v_1, v_1 \rangle \oplus \left( \langle v_1, v_1 \rangle e^{i\psi} \right) \\
&= \left( \langle v_1, v_1 \rangle \oplus \langle v_1, v_1 \rangle \right) e^{i\psi} \\
&= \left( \widetilde{2} \otimes \langle v_1, v_1 \rangle \right) e^{i\psi}.
\end{aligned}$$

*Thus, we get*

$$\widetilde{\sqrt{2}} \otimes \langle v_1, v_1 \rangle = \widetilde{2} \otimes \langle v_1, v_1 \rangle,$$

*which implies $\langle v_1, v_1 \rangle = \tilde{0}$ from the $\alpha$-level sets:*

$$\sqrt{2}\langle v_1, v_1 \rangle_\alpha^- = 2\langle v_1, v_1 \rangle_\alpha^- \qquad and \qquad \sqrt{2}\langle v_1, v_1 \rangle_\alpha^+ = 2\langle v_1, v_1 \rangle_\alpha^+$$

*Therefore, $v_1 = 0$, which is a contraction.*

*5.2. Complex Fuzzy Inner Product Spaces Based on CFNs of Type II*

A complex fuzzy inner product on $X$ defines a fuzzy number

$$\|x\| = \sqrt{\langle x, x \rangle}$$

for all $x \in X$. In fact, $\|x\| = \sqrt{|\langle x, x \rangle|}$ from the the positive-definite property of an inner product.

To begin with, for real numbers $a, b \in \mathbb{R}$, $\widetilde{a + ib} = \tilde{a} \oplus i\,\tilde{b}$. Especially, $\widetilde{\pm i} = i\,\tilde{b}$ with $b = \pm 1$.

**Theorem 2.** *Given a complex fuzzy inner product space $(V, \langle \cdot, \cdot \rangle)$ and for given two element $v, w \in V$, if $\langle v, w \rangle = x \oplus i\,y$ for some fuzzy real numbers $x, y \in F(\mathbb{R})$, then $x = \tilde{a}$ and $y = \tilde{b}$ for some real numbers $a, b \in \mathbb{R}$.*

**Proof.** Note that
$$0 \leq \langle v + w, v + w \rangle = \langle v, v \rangle \oplus \langle v, w \rangle \oplus \langle w, v \rangle \oplus \langle w, w \rangle$$

implies that
$$\langle v, w \rangle \oplus \langle w, v \rangle = (x \oplus x) \oplus i\,(y \ominus y)$$

is a fuzzy real number. Thus $y \ominus y = \tilde{0}$ and so $y = \tilde{b}$ for some $b \in \mathbb{R}$. And, from

$$\langle v, w \rangle \ominus \langle w, v \rangle = (x \ominus x) \oplus i\,(\tilde{2} \otimes y),$$

we get

$$\begin{aligned}
0 \leq \langle iv + w, iv + w \rangle &= \langle iv, iv + w \rangle \oplus \langle w, iv + w \rangle \\
&= \langle v, v \rangle \oplus \tilde{i} \otimes \langle v, w \rangle \oplus \widetilde{-i} \otimes \langle w, v \rangle \oplus \langle w, w \rangle \\
&= (\langle v, v \rangle \oplus \langle w, w \rangle) \oplus \tilde{i} \otimes (\langle v, w \rangle \ominus \langle w, v \rangle) \\
&= (\langle v, v \rangle \oplus \langle w, w \rangle \ominus \tilde{2} \otimes y) \oplus i\,(x \ominus x),
\end{aligned}$$

which implies that
$$x \ominus x = \tilde{0},$$

thus $x = \tilde{a}$ for some $a \in \mathbb{R}$. $\quad\square$

We give a simple application regarding the inner product of complex fuzzy numbers which is of type II.

**Example 6.** *Let $\langle \,, \rangle_0$ be a given inner product of some Hilbert space $X$ over $\mathbb{C}$. If $\langle x, y \rangle$ is given by $\langle x, y \rangle = \tilde{a} \oplus i\,\tilde{b}$ where*
$$\begin{cases} a = Re\langle x, y \rangle_0 \\ b = Im\langle x, y \rangle_0, \end{cases}$$

*that is, (Case I) If $a \neq b$, then*
$$\langle x, y \rangle(t) = \begin{cases} 1, & t = a, \\ i, & t = b, \\ 0, & o.w. \end{cases}$$

*(Case II) If $a = b$, then*

$$\langle x, y \rangle(t) = \begin{cases} 1 + i, & t = a, \\ 0, & o.w. \end{cases}$$

*Then $(IP_1), (IP_2), (IP_4), (IP_5),$ and $(IP_6)$ are clearly holds. For $(IP_3)$,*

$$
\begin{aligned}
\overline{\langle x, y \rangle} &= \widetilde{a} \oplus i(\ominus \widetilde{b}) \\
&= \widetilde{a} \oplus i(\widetilde{-b}) \\
&= \widetilde{Re\langle y, x \rangle_0} \oplus i(\widetilde{Im\langle y, x \rangle_0}) \\
&= \langle y, x \rangle.
\end{aligned}
$$

*Hence $\langle x, y \rangle$ is an inner product of the given Hilbert space X.*

**Remark 3.** *Theorem 2 shows that an inner product complex fuzzy space is trivial. To find more meaningful complex fuzzy space, we will change the condition of positive definiteness $(IP_4)$ in Definition 16 to that of non-degeneracy (see the Section 5).*

**Lemma 4** ([7] (Lemma 3.2)). *A real fuzzy inner product space X together with its corresponding norm $\|\cdot\|$ satisfy the Cauchy-Schwartz inequality*

$$|\langle x, y \rangle| \leq \|x\| \otimes \|y\|$$

*for all $x, y \in X$.*

From now on, the result of Theorem 2 is not used, which enables us to apply the arguments below to the real fuzzy inner product space by letting the imaginary part be $\tilde{0}$.

**Remark 4.** *Given a complex fuzzy number $\eta = x \oplus iy$, where both $x$ and $y$ are fuzzy real numbers, and given $\alpha \in (0, 1]$, the equalities*

$$|\eta|_\alpha^- = \sqrt{(|x|_\alpha^-)^2 + (|y|_\alpha^-)^2} = |\bar{\eta}|_\alpha^-$$

*and*

$$|\eta|_\alpha^+ = \sqrt{(|x|_\alpha^+)^2 + (|y|_\alpha^+)^2} = |\bar{\eta}|_\alpha^+$$

*hold.*

**Remark 5.** *For a vector $v$ in a fuzzy inner product space $(V, \langle \cdot, \cdot \rangle)$ and for $t \geq 0$,*

$$\|v\|(t) = \sqrt{|\langle v, v \rangle|}(t) = |\langle v, v \rangle|(t^2) = (\|v\|^2)(t^2),$$

*which implies $\|v\|_\alpha^- = \sqrt{(\|v\|^2)_\alpha^-}$ and $\|v\|_\alpha^+ = \sqrt{(\|v\|^2)_\alpha^+}$ for each $\alpha \in (0, 1]$.*

The following lemma is easily checked:

**Lemma 5.** *Given a fuzzy real number $x \in F(\mathbb{R})$,*
*(i) if $|x|_\alpha^+ = |x_\alpha^-|$, then $|x|_\alpha^+ = -x_\alpha^-$ and $(\widetilde{|x|_\alpha^+ \otimes x})_\alpha^- = -(|x|_\alpha^+)^2$,*
*(ii) if $|x|_\alpha^+ = |x_\alpha^+|$, then $|x|_\alpha^+ = x_\alpha^+$ and $(\widetilde{-|x|_\alpha^+ \otimes x})_\alpha^- = -(|x|_\alpha^+)^2$.*

**Theorem 3.** *For vectors $v, w$, and for each $\alpha \in (0, 1]$, we have*

$$|\langle v, w \rangle|_\alpha^+ \leq \|v\|_\alpha^- \|w\|_\alpha^-.$$

*Hence, it holds that*    $|\langle v, w \rangle| \leq \|v\| \otimes \|w\|$.

**Proof.** Since all of $|\langle v, w \rangle|$, $\|v\|$ and $\|w\|$ are fuzzy real numbers, it suffices to show that the inequality $|\langle v, w \rangle|_\alpha^+ \leq \|v\|_\alpha^- \|w\|_\alpha^-$ holds for each $\alpha \in (0, 1]$. If $w$ is a zero vector $\vec{0}$, then

$$|\langle v, w \rangle| = |\langle v, \vec{0} \rangle| = |\langle v, 0 \cdot v \rangle| = |\tilde{0} \otimes \langle v, v \rangle| = \tilde{0},$$

which implies $|\langle v, w \rangle|_\alpha^+ = 0$ and so the theorem holds. Assume that $w$ is not a zero vector. Then from the Definition 16 (IP5), $\|w\|_\alpha^- \neq 0$. Denote $\langle v, w \rangle = x \oplus \mathrm{i}y$ by fuzzy real numbers $x$ and $y$. Let

$$a = \begin{cases} \dfrac{|x|_\alpha^+}{(\|w\|_\alpha^-)^2}, & \text{if } |x|_\alpha^+ = |x_\alpha^-|, \\[2mm] \dfrac{-|x|_\alpha^+}{(\|w\|_\alpha^-)^2}, & \text{if } |x|_\alpha^+ = |x_\alpha^+|. \end{cases}$$

and put $b$, related to $y_\alpha^-$ and $y_\alpha^+$, in a similar way. Consider $\gamma = a + ib \in \mathbb{C}$. Then, the inequality

$$\begin{aligned} \tilde{0} &\leq |\langle v + \gamma w, v + \gamma w \rangle| = \langle v + \gamma w, v + \gamma w \rangle \\ &= \|v\|^2 \oplus \tilde{\bar{\gamma}} \otimes \langle v, w \rangle \oplus \tilde{\gamma} \otimes \overline{\langle v, w \rangle} \oplus |\tilde{\gamma}|^2 \otimes \|w\|^2 \end{aligned}$$

holds and $\tilde{\bar{\gamma}} \otimes \langle v, w \rangle \oplus \tilde{\gamma} \otimes \overline{\langle v, w \rangle}$ becomes a fuzzy real number. Thus we can rewrite the equality as follows:

$$\langle v + \gamma w, v + \gamma w \rangle = \|v\|^2 \oplus \left( \tilde{2} \otimes \tilde{a} \otimes x \oplus \widetilde{a^2} \otimes \|w\|^2 \right) \oplus \left( \tilde{2} \otimes \tilde{b} \otimes y \oplus \widetilde{b^2} \otimes \|w\|^2 \right).$$

From Lemma 5,

$$\begin{aligned} \left( \tilde{2} \otimes \tilde{a} \otimes x \oplus \widetilde{a^2} \otimes \|w\|^2 \right)_\alpha^- &= 2 \left( \tilde{a} \otimes x \right)_\alpha^- + a^2 \left( \|w\|_\alpha^- \right)^2 \\ &= \frac{-2 (|x|_\alpha^+)^2}{(\|w\|_\alpha^-)^2} + \frac{(|x|_\alpha^+)^2}{(\|w\|_\alpha^-)^2} \\ &= \frac{-(|x|_\alpha^+)^2}{(\|w\|_\alpha^-)^2}. \end{aligned}$$

Similarly, we get

$$\left( \tilde{2} \otimes \tilde{b} \otimes y \oplus \widetilde{b^2} \otimes \|w\|^2 \right)_\alpha^- = \frac{-(|y|_\alpha^+)^2}{(\|w\|_\alpha^-)^2},$$

which gives

$$0 \leq \left( \langle v + \gamma w, v + \gamma w \rangle \right)_\alpha^- = (\|v\|_\alpha^-)^2 - \frac{(|x|_\alpha^+)^2 + (|y|_\alpha^+)^2}{(\|w\|_\alpha^-)^2} = (\|v\|_\alpha^-)^2 - \frac{(|\langle v, w \rangle|_\alpha^+)^2}{(\|w\|_\alpha^-)^2}$$

and

$$|\langle v, w \rangle|_\alpha^+ \leq \|v\|_\alpha^- \|w\|_\alpha^-.$$

$\square$

## 6. Complex Fuzzy Scalar Product

We already saw that no inner product can exist on the complex fuzzy numbers of type I. In Linear Algebra, recall the concept of a scalar product, which is a weaker version of the concept on an inner product. We introduce a complex fuzzy scalar product for a generalization of a complex fuzzy inner product. In this section, $CF(\mathbb{R})$ does not restrict to the case of type II, except Theorem 4.

**Definition 17.** *Let $X$ be a vector space over $\mathbb{C}$. A complex-valued fuzzy scalar product on $X$ is a mapping $\langle\!\langle \, , \, \rangle\!\rangle : X \times X \to CF(\mathbb{R})$ such that for all vectors $x, y, z \in X$ ans $r \in \mathbb{C}$, we have*

$(SP_1)$ $\langle\!\langle x+y,z \rangle\!\rangle = \langle\!\langle x,z \rangle\!\rangle \oplus \langle\!\langle y,z \rangle\!\rangle$,

$(SP_2)$ $\langle\!\langle rx,y \rangle\!\rangle = \widetilde{r} \otimes \langle\!\langle x,y \rangle\!\rangle$,

$(SP_3)$ $\langle\!\langle x,y \rangle\!\rangle = \overline{\langle\!\langle y,x \rangle\!\rangle}$,

$(SP_4(1))$     $|\langle\!\langle v,w \rangle\!\rangle|(st) \leq \max\left(\sqrt{|\langle\!\langle v,v \rangle\!\rangle|}(s), \sqrt{|\langle\!\langle w,w \rangle\!\rangle|}(t)\right)$
*whenever* $s \geq (\sqrt{|\langle\!\langle v,v \rangle\!\rangle|})_1^-$, $t \geq (\sqrt{|\langle\!\langle w,w \rangle\!\rangle|})_1^-$ *and* $st \geq |\langle\!\langle v,w \rangle\!\rangle|_1^-$.

$(SP_4(2))$     $|\langle\!\langle v,w \rangle\!\rangle|(st) \geq \min\left(\sqrt{|\langle\!\langle v,v \rangle\!\rangle|}(s), \sqrt{|\langle\!\langle w,w \rangle\!\rangle|}(t)\right)$
*whenever* $s \leq (\sqrt{|\langle\!\langle v,v \rangle\!\rangle|})_1^-$, $t \leq (\sqrt{|\langle\!\langle w,w \rangle\!\rangle|})_1^-$ *and* $st \leq |\langle\!\langle v,w \rangle\!\rangle|_1^-$.

The vector space $X$ with a complex-valued fuzzy scalar product is called a complex fuzzy scalar product space.

**Theorem 4.** *Any complex-valued fuzzy inner product,* $\langle \cdot, \cdot \rangle$, *on a vector space $X$ over $\mathbb{C}$ is a complex-valued fuzzy scalar product.*

**Proof.** It suffices to show that both $SP_4(1)$ and $SP_4(2)$ hold. To begin with, recall that $\|v\| = \sqrt{|\langle v,v \rangle|}$. Let $\alpha = |\langle v,w \rangle|(st)$. Note that if both $s$ and $t$ are nonnegative, then Theorem 3 gives

$$0 \leq |\langle v,w \rangle|_\alpha^- \leq st \leq |\langle v,w \rangle|_\alpha^+ \leq (\sqrt{|\langle v,v \rangle|})_\alpha^- (\sqrt{|\langle w,w \rangle|})_\alpha^-,$$

so we get either $0 \leq s \leq (\sqrt{|\langle v,v \rangle|})_\alpha^-$ or $0 \leq t \leq (\sqrt{|\langle w,w \rangle|})_\alpha^-$ holds.

To show $SP_4(1)$, consider its hypothesis, $s \geq (\sqrt{|\langle v,v \rangle|})_1^-$, $t \geq (\sqrt{|\langle w,w \rangle|})_1^-$ and $st \geq |\langle v,w \rangle|_-^1$, which implies either

$$0 \leq (\sqrt{|\langle v,v \rangle|})_1^- \leq s \leq (\sqrt{|\langle v,v \rangle|})_\alpha^- \leq (\sqrt{|\langle v,v \rangle|})_1^-$$

or

$$0 \leq (\sqrt{|\langle w,w \rangle|})_1^- \leq t \leq (\sqrt{|\langle w,w \rangle|})_\alpha^- \leq (\sqrt{|\langle w,w \rangle|})_1^-$$

holds, in other words, either $s = (\sqrt{|\langle v,v \rangle|})_1^-$ or $t = (\sqrt{|\langle w,w \rangle|})_1^-$. Thus we get either $\sqrt{|\langle v,v \rangle|}(s) = 1$ or $\sqrt{|\langle w,w \rangle|}(t) = 1$, and obtain the inequality in $SP_4(1)$.

To show $SP_4(2)$, consider its hypothesis, $s \leq (\sqrt{|\langle v,v \rangle|})_-^1$, $t \leq (\sqrt{|\langle w,w \rangle|})_-^1$ and $st \leq |\langle v,w \rangle|_-^1$, which implies either

$$s \leq (\sqrt{|\langle v,v \rangle|})_\alpha^- \leq (\sqrt{|\langle v,v \rangle|})_1^-$$

or

$$t \leq (\sqrt{|\langle w,w \rangle|})_\alpha^- \leq (\sqrt{|\langle w,w \rangle|})_1^-$$

holds (even in case that either $s$ or $t$ is negative). Thus we get either $\sqrt{|\langle v,v \rangle|}(s) \leq \alpha$ or $\sqrt{|\langle w,w \rangle|}(t) \leq \alpha$, and obtain the inequality in $SP_4(2)$. $\square$

**Remark 6.** *Theorem 4 is meaningful only in case of type II. See Section 5.1.*

**Lemma 6.** *$(SP_1)$ is equivalent to*

$$|\langle\!\langle v,w \rangle\!\rangle|_\alpha^+ \leq (\sqrt{|\langle\!\langle v,v \rangle\!\rangle|})_\alpha^+ \cdot (\sqrt{|\langle\!\langle w,w \rangle\!\rangle|})_\alpha^+, \quad \alpha \in (0,1]. \tag{7}$$

**Proof.** Assume the Equation (7) holds. Given $s$ and $t$ satisfying the hypothesis of $(SP_1)$, let $\alpha = |\langle\!\langle v,w \rangle\!\rangle|(st)$. Then,

$$st \leq |\langle\!\langle v,w \rangle\!\rangle|_\alpha^+ \leq (\sqrt{|\langle\!\langle v,v \rangle\!\rangle|})_\alpha^+ \cdot (\sqrt{|\langle\!\langle w,w \rangle\!\rangle|})_\alpha^+$$

from the Equation (7). The convexity and the hypothesis of ($SP_1$) say that either

$$0 \leq (\sqrt{|\langle\langle v,v\rangle\rangle|})_\alpha^- \leq (\sqrt{|\langle\langle v,v\rangle\rangle|})_1^- \leq s \leq (\sqrt{|\langle\langle v,v\rangle\rangle|})_\alpha^+$$

or

$$0 \leq (\sqrt{|\langle\langle w,w\rangle\rangle|})_\alpha^- \leq (\sqrt{|\langle\langle v,v\rangle\rangle|})_1^- \leq t \leq (\sqrt{|\langle\langle w,w\rangle\rangle|})_\alpha^+,$$

which implies that, at least, one of $\sqrt{|\langle\langle v,v\rangle\rangle|}(s)$ and $\sqrt{|\langle\langle w,w\rangle\rangle|}(t)$ is bigger than or equal to $\alpha$. Therefore,

$$|\langle\langle v,w\rangle\rangle|(st) = \alpha \leq \max\left(\sqrt{|\langle\langle v,v\rangle\rangle|}(s), \sqrt{|\langle\langle w,w\rangle\rangle|}(t)\right).$$

Conversely, suppose that ($SP_1$) holds. To show by contradiction, assume that there exist a number $\alpha \in (0,1]$ and vectors $v$ and $w$ such that

$$|\langle\langle v,w\rangle\rangle|_\alpha^+ > (\sqrt{|\langle\langle v,v\rangle\rangle|})_\alpha^+ \cdot (\sqrt{|\langle\langle w,w\rangle\rangle|})_\alpha^+.$$

Then we can find two positive numbers $s$ and $t$ satisfying

$$s > (\sqrt{|\langle\langle v,v\rangle\rangle|})_\alpha^+ \geq (\sqrt{|\langle\langle v,v\rangle\rangle|})_1^-,$$

$$t > (\sqrt{|\langle\langle w,w\rangle\rangle|})_\alpha^- \geq (\sqrt{|\langle\langle w,w\rangle\rangle|})_1^-$$

and

$$st = (\sqrt{|\langle\langle v,w\rangle\rangle|})_\alpha^- \geq |\langle\langle v,w\rangle\rangle|_1^-.$$

Then, both of $\sqrt{|\langle\langle v,v\rangle\rangle|}(s)$ and $\sqrt{|\langle\langle w,w\rangle\rangle|}(t)$ is less than $\alpha$, which, together with ($SP_1$), gives

$$\alpha = |\langle\langle v,w\rangle\rangle|(st) \leq \max\left(\sqrt{|\langle\langle v,v\rangle\rangle|}(s), \sqrt{|\langle\langle w,w\rangle\rangle|}(t)\right) < \alpha.$$

It is a contradiction. □

**Lemma 7.** *($SP_2$) is equivalent to*

$$|\langle\langle v,w\rangle\rangle|_\alpha^- \leq (\sqrt{|\langle\langle v,v\rangle\rangle|})_\alpha^- \cdot (\sqrt{|\langle\langle w,w\rangle\rangle|})_\alpha^-, \quad \alpha \in (0,1]. \tag{8}$$

**Proof.** Assume the Equation (8) holds. Given $s$ and $t$ satisfying the hypothesis of ($SP_2$), the conclusion of ($SP_2$) holds trivially if one of $\sqrt{|\langle\langle v,v\rangle\rangle|}(s)$ or $\sqrt{|\langle\langle w,w\rangle\rangle|}(t)$ is zero. So assume that both of them are greater than zero. Let $0 < \alpha = \sqrt{|\langle\langle v,v\rangle\rangle|}(s)$, $0 < \beta = \sqrt{|\langle\langle w,w\rangle\rangle|}(t)$ and $\gamma = \min(\alpha,\beta)$. Then, both $(\sqrt{|\langle\langle v,v\rangle\rangle|})_\alpha^- \leq s$ and $(\sqrt{|\langle\langle w,w\rangle\rangle|})_\alpha^- \leq t$ hold. Thus, the Equation (8) and the nondecreasing property of $\delta \mapsto (\sqrt{|\langle\langle \cdot,\cdot\rangle\rangle|})_\delta^-$ give

$$\begin{aligned}
|\langle\langle v,w\rangle\rangle|_\gamma^- &\leq (\sqrt{|\langle\langle v,v\rangle\rangle|})_\gamma^- \cdot (\sqrt{|\langle\langle w,w\rangle\rangle|})_\gamma^- \\
&\leq (\sqrt{|\langle\langle v,v\rangle\rangle|})_\alpha^- \cdot (\sqrt{|\langle\langle w,w\rangle\rangle|})_\beta^- \\
&\leq st \\
&\leq |\langle\langle v,w\rangle\rangle|_1^-,
\end{aligned}$$

which, together with the convexity property, implies

$$|\langle\langle v,w\rangle\rangle|(st) \geq \gamma = \min(\alpha,\beta) = \min\left(\sqrt{|\langle\langle v,v\rangle\rangle|}(s), \sqrt{|\langle\langle w,w\rangle\rangle|}(t)\right).$$

Conversely, suppose that ($SP_2$) holds. Let $\alpha \in (0,1]$ be given.

Case (1) $(\sqrt{|\langle\!\langle v,v\rangle\!\rangle|})_{\alpha}^{-} \cdot (\sqrt{|\langle\!\langle w,w\rangle\!\rangle|})_{\alpha}^{-} \leq |\langle\!\langle v,w\rangle\!\rangle|_{1}^{-}$

For $s = (\sqrt{|\langle\!\langle v,v\rangle\!\rangle|})_{\alpha}^{-}$ and $t = (\sqrt{|\langle\!\langle w,w\rangle\!\rangle|})_{\alpha}^{-}$, both $\sqrt{|\langle\!\langle v,v\rangle\!\rangle|}(s)$ and $\sqrt{|\langle\!\langle w,w\rangle\!\rangle|}(t)$ are greater than or equal to $\alpha$, thus ($SP_2$) says that

$$|\langle\!\langle v,w\rangle\!\rangle|(st) \geq \min\left(\sqrt{|\langle\!\langle v,v\rangle\!\rangle|}(s), \sqrt{|\langle\!\langle w,w\rangle\!\rangle|}(t)\right) \geq \alpha.$$

Therefore,

$$|\langle\!\langle v,w\rangle\!\rangle|_{\alpha}^{-} \leq st = (\sqrt{|\langle\!\langle v,v\rangle\!\rangle|})_{\alpha}^{-} \cdot (\sqrt{|\langle\!\langle w,w\rangle\!\rangle|})_{\alpha}^{-}.$$

Case (2) $(\sqrt{|\langle\!\langle v,v\rangle\!\rangle|})_{\alpha}^{-} \cdot (\sqrt{|\langle\!\langle w,w\rangle\!\rangle|})_{\alpha}^{-} > |\langle\!\langle v,w\rangle\!\rangle|_{1}^{-}$

The Equation (8) holds since

$$|\langle\!\langle v,w\rangle\!\rangle|_{\alpha}^{-} \leq |\langle\!\langle v,w\rangle\!\rangle|_{1}^{-} \leq (\sqrt{|\langle\!\langle v,v\rangle\!\rangle|})_{\alpha}^{-} \cdot (\sqrt{|\langle\!\langle w,w\rangle\!\rangle|})_{\alpha}^{-}.$$

□

A complex fuzzy scalar product on $X$ defines a fuzzy number

$$\|x\| = \sqrt{|\langle\!\langle x,x\rangle\!\rangle|}$$

for all $x \in X$. Lemmas 6 and 7 imply that

**Theorem 5.** *For vectors $x, y$, we have the following inequalities*

$$\begin{cases} \text{(i) } |\langle\!\langle x,y\rangle\!\rangle| \leq \|x\| \otimes \|y\|. \\ \text{(ii) } \|x+y\| \leq \|x\| \oplus \|y\|. \end{cases}$$

**Proof.** (ii) By the definition and (i), we have

$$\begin{aligned} \|x+y\|^2 &= |\langle\!\langle x+y, x+y\rangle\!\rangle| \\ &\leq \|x\|^2 \oplus |\langle\!\langle x,y\rangle\!\rangle| \oplus |\langle\!\langle x,y\rangle\!\rangle| \oplus \|y\|^2 \\ &\leq \|x\|^2 \oplus \|x\| \otimes \|y\| \oplus \|y\| \otimes \|x\| \oplus \|y\|^2 \\ &= (\|x\| \oplus \|y\|)^2. \end{aligned}$$

□

## 7. Conclusions

In this study, we defined complex fuzzy numbers (CFNs) of the Cartesian form and the polar form and took a look at the advantage for each type, providing examples. To study them, we proposed some operations and properties such as triangular inequality in fuzzy real number set. Based on proposed definitions and properties, complex fuzzy inner product space has been proposed based on CFN of the Cartesian form. Especially, we made a more clear definition of modulus, which gave an easy proof for Cauchy-Schwartz inequality. We also showed the non-existence of the inner product on CFNs of the polar form. This implies that the polar form cannot substitute Cartesian form, so we need to apply each type in the right situation. And we proposed a new concept, called complex fuzzy scalar product, and proved its some basic properties such as both Cauchy-Schwartz inequality and triangular inequality. Regarding this, the condition for the existence of a scalar product on CFN of type I will be discussed in our further study.

**Author Contributions:** The individual contributions of the authors are as follows: conceptualization, T.B., J.E.L., K.Y.L., and J.H.Y.; methodology, T.B., J.E.L., K.Y.L., and J.H.Y.; writing—original draft preparation, T.B., J.E.L.; writing—review and editing, J.E.L. All authors have read and agreed to the published version of the manuscript.

**Funding:** Jin Hee Yoon was supported by the National Research Foundation of Korea (NRF) grant funded by the Korea government (MSIT) (No. 2020R1A2C1A01011131). Taechang Byun was supported by Basic Science Research Program through the National Research Foundation of Korea (NRF) funded by the Ministry of Education (2018R1D1A1A02047995). Ji Eun Lee was supported by the National Research Foundation of Korea (NRF) grant funded by the Korea government (MSIT) (2019R1A2C1002653). Keun Young Lee was supported by NRF-2017R1C1B5017026 funded by the Korean Government.

**Acknowledgments:** The authors wish to thank the referees for their invaluable comments on the original draft.

**Conflicts of Interest:** The authors declare no conflict of interest.

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
