# Peer review of "Classification of Complex Fuzzy Numbers and Fuzzy Inner Products"

_mathematics, doi:10.3390/math8091626_

Round 1
Reviewer 1 Report
1. Section 3 is just the easy consequence of Zadeh's extension principle. The motivation for considering the square root of fuzzy number is not clear. Why we need to consider the square root of fuzzy numbers. Also, the definition of square of fuzzy complex number is given in Definition 4.8
2. Regarding the fuzzy complex numbers, the comparisons between Definitions 4.1 and 4.4 should be provided. The advantage and disadvantage should be provided.
3. The authors need to provide some simple applications regarding the inner product of complex fuzzy numbers.
Author Response
- Thank you for your valuable advice. Based on your comments, the whole section 3 has been removed. But we need Definition 4.8 (Definition 3.8 in revised version) throughout sections 5 and 6.
2. Section 4 –> Section 3 (in revised version)
Definition 3.1 is for the complex fuzzy set S with polar form. (a general version). But Definition 3.4 is for the complex fuzzy number.
Actually, a fuzzy number is a special kind of a fuzzy set.
For this, please see following links:
1) Definition of a fuzzy set : https://en.wikipedia.org/wiki/Fuzzy_set
2) Definition of a fuzzy number : https://en.wikipedia.org/wiki/Fuzzy_number
3. Thank you for your comments. We provide some examples regarding inner products (Example 5.2, 5.4) In addition, Example 6.2, which is for complex fuzzy scalar product has been inserted in our revised version.
Reviewer 2 Report
The paper ‘Classification of complex fuzzy numbers and fuzzy inner products’ is a very interesting paper. It is well developed with a strong theoretical knowledge base. It can also contribute to extending the knowledge on complex fuzzy numbers, and fuzzy inner products. However, there is a need for significant clarifications and improvements to make it publishable. The authors can improve paper based on the following suggestions.
The abstract is inadequate and does not provide any background and relevance to the study. Moreover, it does not provide any information about the critical findings, implications, and contributions to knowledge. The introduction should also adequately provide the background, research gap, objective and relevance of the study.
Similarly, the conclusion is not properly structured and lacks information about the critical findings, significance, and implication of the study, and the contribution the paper has made. Also, it does not provide any discussion on findings from the study vis-a-vis the findings from the existing literature, if any available. In the absence of which, it appears that the study is very novel and standalone, and therefore requires to be validated before being accepted.
The authors also need to explain the critical aspects of complex fuzzy numbers, fuzzy complex numbers, fuzzy inner products, complex fuzzy scalar products, etc. These aspects should be articulated more clearly and lucidly to make it useful to a more general reader and users than the limited specialists in the field.
Nevertheless, one of the most important aspects the authors should look into is the validity of their findings. The definitions, they have proposed, the classification and products they have made should be validated adequately either by theoretical proofs or evidence from literature, without which they may remain as propositions.
The other important aspect is that despite being a purely theoretical study the authors need to explain how the study will be useful in different fields of study, for example in modeling for decision making or any Artificial Intelligence applications.
Moreover, even though the derivations and proofs are well developed, the paper lacks a good linkage between different sections. The writing is also at times a bit weird and needs to be improved to make it more readable and scientific.
Author Response
In classical complex analysis, we have two different types of notations in the complex number set. One is Cartesian form (z=a+bi) and the other one is the polar form. Even though we have Cartesian form, the polar form is needed depending on the situation. The Cartesian form is generally convenient but if we have to use the angle such as in triangular functions, the polar form is much better.
Likewise, in fuzzy complex analysis , we can first consider the Cartesian complex fuzzy number form, but depending on the situation it is needed to define polar form to express some special cases.
For example, if we want to express some situation with “periodic” time series which has fuzzy amplitude such as Example 3.3, the polar form is much better than the Cartesian form. Polar form enables us to express the periodicity easily like in Example 3.3, if the data includes the monthly property or the seasonality.
However, in the general situations, Catesian form is easy to use. Thus we need to study both of them. And, the difference of those two types emerges in the study of the complex fuzzy inner product. The complex fuzzy inner product does not exist based on the polar form. Thus we need to consider both of those two types and apply them in the appropriate situation. And we also showed that the Cauchy-Schwartz inequality holds in the Cartesian form in the complex fuzzy number set.
Reviewer 3 Report
It is necessary to increase the volume of abstract.
Add sources of literature (references).
The article is interesting. the authors described the solution to the problem well enough.
Author Response
Thank you for your valuable advice. We give an more detailed explnation in the abstract and in the main text. And we add sources of references more clearly.
Round 2
Reviewer 2 Report
The authors have clarified the reasons of use of the cartesian and polar form of complex fuzzy numbers. They have improved the paper by adding an example 5.6.However, some of the concerns still remain. The authors should look into the following aspects.
The abstract and conclusion should be improved.
Explanation on the validity and significance of the findings should be given.
Author Response
Please see the attatchment.
